# Gut-Brain Axis: Role of Microbiome, Metabolomics, Hormones, and Stress in Mental Health Disorders

**DOI:** 10.3390/cells13171436

**Published:** 2024-08-27

**Authors:** Ankita Verma, Sabra S. Inslicht, Aditi Bhargava

**Affiliations:** 1Department of Obstetrics, Gynecology & Reproductive Sciences, University of California at San Francisco, 513 Parnassus Avenue, San Francisco, CA 94143, USA; ankitaverma7798@gmail.com; 2San Francisco VA Health Care System, San Francisco, CA 94121, USA; sabra.inslicht@ucsf.edu; 3Department of Psychiatry and Behavioral Sciences, University of California at San Francisco, San Francisco, CA 94143, USA

**Keywords:** depression, metabolomics, placebo, PTSD, sex differences

## Abstract

The influence of gut microbiome, metabolites, omics, hormones, and stress on general and mental health is increasingly being recognized. Ancient cultures recognized the importance of diet and gut health on the overall health of an individual. Western science and modern scientific methods are beginning to unravel the foundations and mechanisms behind some of the ancient beliefs and customs. The gut microbiome, an organ itself, is now thought to influence almost all other organs, ranging from the brain to the reproductive systems. Gut microbiome, metabolites, hormones, and biological sex also influence a myriad of health conditions that range from mental health disorders, obesity, gastrointestinal disorders, and cardiovascular diseases to reproductive health. Here, we review the history and current understanding of the gut–brain axis bidirectional talk in various mental health disorders with special emphasis on anxiety and depressive disorders, whose prevalence has increased by over 50% in the past three decades with COVID-19 pandemic being the biggest risk factor in the last few years. The vagal nerve is an important contributor to this bidirectional talk, but other pathways also contribute, and most remain understudied. Probiotics containing *Lactobacillus* and *Bifidobacterium* species seem to have the most impact on improvement in mental health symptoms, but the challenge appears to be maintaining sustained levels, especially since neither *Lactobacillus* nor *Bifidobacterium* can permanently colonize the gut. Ancient endogenous retroviral DNA in the human genome is also linked to several psychiatric disorders, including depression. These discoveries reveal the complex and intricately intertwined nature of gut health with mental health disorders.

## 1. Introduction

The concept of a connection between digestion, emotions, and behavior has roots in ancient Chinese, Greek, and Indian traditional medicines [1]. This idea gained traction as early as the 18th century when discussions on how the microbiome influences the mind in a more systemic manner began to emerge. In 1825, the French gastronome and epicurean Jean Anthelme Brillat-Savarin published his masterpiece book Physiologie du goût (The Physiology of Taste) with the famous quote: “Dis-moi ce que tu manges, je te dirai ce que tu es” (“Tell me what you eat, and I will tell you who you are”). With the advancement of research and rapidly evolving technology [1], the concept of the “gut–brain axis” has emerged and solidified. In the past decade, large-scale studies like Metagenomics of the Human Intestinal Tract (MetaHIT), next-generation sequencing, and “omics” platforms have all provided essential data on biomarkers and microbiota composition [2]. The development of sophisticated computational platforms along with integrated systems data analyses has led to a deeper and better understanding of the gut–brain crosstalk [3,4] and the idiom “you are what you eat”. Our microbiome functions like an organ, providing vital nutrients and factors essential for the overall growth of cardiovascular, neurological, and psychological development [5].

The enteric nervous system, part of the autonomic nervous system, contained in the gut, is also referred to as the “small brain” and releases a range of hormones and neurotransmitters to regulate the motor, sensory, absorptive, and secretory functions of the gastrointestinal tract (GI). Conversely, the brain influences GI function via the release of neurotransmitters and hormones to the autonomic nervous system nerve endings in the gut; this crosstalk is referred to as the gut–brain bidirectional axis [6]. This crosstalk is also a double-edged sword: our microbiome manifests both beneficial and harmful effects on human physiology. Microbiome dysbiosis, or alteration in the normal composition of the microbiome, is associated with many health conditions, including GI and mental health disorders.

Mood and anxiety disorders together are the most common mental health disorders globally, with over 50% increase in prevalence in the past three decades [7]. According to WHO, depressive disorders are prevalent in 3.8% to 10% of the population, with differences in prevalence between men and women. Additionally, more than 10% of females experience postpartum depression, highlighting the significant sex disparity in depressive conditions. A cross-sectional study showed that about 37% (*n* = 473) of adolescents suffer from depression [8], and more than 700,000 people die by suicide every year [9]. The impact of mental health disorders on health is exacerbated by poor diet, processed food intake, alcohol and substance use, poor sleep, and social isolation. For this reason, it is of paramount importance to address the rising rates of depression. Microbiome restoration remains a non-invasive therapeutic modality to treat mental health disorders, including depression and anxiety.

This review is not an exhaustive review of the gut–brain axis or all neurological and mental health disorders in which the microbiome is implicated. Here, we discuss how the gut microbiome, metabolites, and biological sex influence the gut–brain axis in a bidirectional manner in health and in pathophysiology. Newer technologies have led to the identification of metabolites produced by gut microbiota as well as various metabolomic components (primary amines, lipids, steroid metabolites). The role of various omics is also discussed in this review. This review addresses the relationship between microbiome and mood disorders, with a particular emphasis on depression and anxiety disorders. The subsequent sections delve into the pathophysiology of depression and anxiety disorders, exploring potential connections between the microbiota, metabolites, and the manifestation of symptoms. Current treatment recommendations for depression are then explored, accompanied by additional advice for patients. The review concludes by scrutinizing limitations in existing studies and their implications for the field of medicine.

## 2. Materials and Methods

We searched the PubMed and bioRxiv databases accessed through July 2024. A mix of keyword and subject heading searches was utilized. Terms associated with mental health conditions like depression, mood, anxiety, stress, PTSD, neuropsychiatric, vagal nerve, quorum sensing, metabolomics, proteomics, inflammatory bowel disease, nutritional psychiatry, prebiotics, probiotics, postbiotics, symptoms of depression, constipation, diarrhea, fecal microbiota transplant, adjunctive therapy, sex differences, were combined with keywords in the search engine query like microbiome, microbiota, gut–brain, metabolites, diet, dysbiosis, gut–brain axis. Both original articles and reviews were used to prepare this review article. Articles older than three decades were included for historical purposes. When possible, original studies were cited, but when conclusions from many studies were included, review articles were cited.

## 3. The Gut

### 3.1. The Gut Microbiome

The microbiome is a complex assembly of multiple organisms present in the human body whose impact on human health and disease is being actively investigated [10]. The human microbiome consists of 10 to 100 trillion different microbial cells [11] with at least 150–400 different types of species. To illustrate the sheer density of organisms in the gut, each square millimeter of the colonic wall harbors 10^11^ cells [10], whereas the small intestinal wall harbors 10^8^ cells. This colonization of the human body starts during the process of birth when the infant passes through the vaginal canal, and the mother’s flora colonizes the infant’s gut [4,12,13]. Over time, this flora changes as the infant starts other foods besides breast milk. Around the age of 2, a child’s gut flora begins to diverge from that of the mother, setting the foundation for the microbial composition the child will have as an adult [10].

#### 3.1.1. Impact of External Stressors on the Microbiome

An individual’s microbiome is a dynamic organ that evolves constantly depending on various factors, including nutrition, stress, exercise, geographical location, age, sex, hormonal levels, and inherited genes [12,13]. External factors such as smoking, alcohol, and drug use are linked to shaping the microbiome. Cigarette smoking influences the microbiome through mechanisms such as altering immune homeostasis, promoting biofilm formation, changing oxygen levels, or through direct contact with the microbes present in cigarettes [14]. Sapkota et al. discovered a wide range of bacterial species in cigarettes that are introduced from tobacco leaves, including soil microorganisms, commensals, and potential human pathogens such as *Acinetobacter*, *Bacillus*, *Burkholderia*, *Clostridium*, *Klebsiella*, and *Pseudomonas aeruginosa* [14]. Another study by Shanahan et al. found that current smokers had a decreased relative abundance of *Prevotella* and *Neisseria* species and an increased abundance of *Firmicutes*, particularly *Streptococcus* and *Veillonella* species, along with *Rothia* (*Actinobacteria*) in the upper GI tract compared to non-smokers [15]. The changes in *Neisseria*, *Streptococcus*, and *Rothia* in smokers suggest an impact from altered oxygen levels. Additionally, reduced duodenal bicarbonate secretion and lower pH in smokers may create selective pressure, favoring acid-tolerant bacteria like *Streptococcus* and *Rothia* over acid-sensitive *Neisseria*. Alcohol use, whether acute or chronic, can lead to gut dysbiosis. In cases of short-term use, this dysbiosis can be partially reversed with abstinence [16]. However, chronic heavy drinkers experience further alterations in their microbiome, including a decrease in *Bifidobacterium*, *Clostridiales*, *Lachnospiraceae*, and *Ruminococcaceae*, all of which are known to produce short-chain fatty acids (SCFAs) [16]. This dysbiosis caused by binge drinking and chronic alcohol use can further intensify cravings and exacerbate psychological conditions such as anxiety and depression. Over half of chronic alcohol abusers [17] and drug abusers (including cocaine and opiates) [18] exhibit intestinal barrier dysfunction or “leaky” gut and dysbiosis.

Immune cytokines play a role in modulating behavior, and levels of pro-inflammatory cytokines like IL-1β, TNF-α, and IL-8 increase in response to various substances of abuse [18]. This pro-inflammatory reaction underscores the interaction between gut microbiota and the immune system in the pathology of substance use disorders. Alcohol and opiate use have been observed to increase pro-inflammatory *Proteobacteria* in the gut. As a result, probiotics have gained interest as a potential therapy and may also be effective in treating anxiety and depression, common co-morbidities of substance use disorder [18]. Another study by Martinez et al. found that the abundances of the *Lachnospira* and *Oscillospira genera*, as well as *Bifidobacterium* species, were significantly higher in cocaine users with HIV compared to HIV-positive individuals who did not use cocaine [19]. These findings show that cocaine use negatively affects the host gut microbiome and its metabolites.

Additionally, both antibiotic and non-antibiotic drugs impact microbiota composition and drug metabolism, with recent findings linking postpartum antibiotic use to depressive symptoms underscoring the need for further clinical studies [20].

#### 3.1.2. Our Microbiome—More than Just Bacteria

The microbiome was thought to be solely bacterial in composition; however, this is not the case. It harbors more than just bacterial species, including viruses, fungi, and even protozoa and helminths [21]. Some key bacterial families that colonize the human gut are *Bacteroidetes*, *Firmicuites*, *Actinobacteria*, *Fusobacteria*, and *Proteobacteria* [12,21]. These bacterial species metabolize complex foods to produce important metabolites like SCFAs, essential vitamins and amino acids, trimethylamine-n-oxides, and lipopolysaccharides (LPS). More specifically, *Candida*, *Escherichia*, *Enterococcus*, and *Streptococcus* belong to serotonin-producing bacteria, *Bifidobacterium* and *Lactobacillus* to gamma-aminobutyric acid (GABA), whereas *Bacillus* and *Serratia* to dopamine by influencing the production of amino acids in the gut from which these hormones/neurotransmitters are derived [22]. Bacterial metabolites also consist of neuromodulators, uremic toxins, pro-inflammatory and anti-inflammatory agents, and metabolites that contribute to the host’s cellular metabolism for energy [22]. Fungi like *Sacchoromyces boulardii* contribute to the production of norepinephrine. Altogether, the cross-talk between the gut and brain allows it to function as a unit through endocrine, immunological, and neuronal signaling pathways [22].

#### 3.1.3. Quorum Sensing and the Microbiome

Quorum sensing is a widespread communication mechanism used by bacteria to coordinate collective behaviors, influencing the structure and function of polymicrobial communities [23]. Quorum sensing induces changes in global gene expression by responding to various autoinducers secreted by bacteria or even the host [23,24,25]. Host factors can also influence quorum-sensing signaling, thereby modulating the outcome of pathogen invasion. For instance, chronic wounds are often infected with both *S. aureus* and *P. aeruginosa*. Interestingly, *P. aeruginosa* easily eliminates *S. aureus* when co-cultured under standard laboratory conditions; the two species coexist and display synergistic antibiotic tolerance in chronic wounds [25]. The quorum-sensing signaling systems have been thoroughly studied in individual bacteria, revealing their involvement in processes such as bacterial colonization, virulence expression, and biofilm formation [25,26]. However, understanding how these systems interact to create an effective bacterial communication network and maintain community homeostasis in multispecies consortia, such as the human gut microflora, remains a challenge. There is growing evidence that chemical communication between species and even across different kingdoms influences the composition of the gut microbiota [27]. A study examining the impact of the quorum sensing on gut microbiota after antibiotic-induced dysbiosis in mice found that autoinducer-2 mediated inter-species communication encourages the growth of *Firmicutes* over *Bacteroidetes* [27]. This shows that autoinducer-2 levels affect the abundance of the major phyla in the gut microbiota, whose balance is crucial for human health [27].

### 3.2. The Intestinal Barrier, Blood Brain Barrier, and Barrier Permeability

Due to the presence of the blood-brain-barrier (BBB), it is improbable that gut neurotransmitters have direct access to the brain [28]. The BBB is composed of 3 layers, the inner layer being the endothelial cells that line the capillary walls in the brain and are connected by tight junctions that impede the movement of substances through the space between adjacent endothelial cells [29,30]. This is known as the paracellular passage [22,23]. The middle layer, specifically the basement membrane, contains peripheral cells and the matrix [23,24,25]. The outer layer is comprised of the extracellular matrix and astrocytes [23,24,25]. Besides being a physical barrier, the BBB incorporates a biochemical component featuring numerous transporters and enzymes. The substantial resistance among the brain microvessel endothelial cells establishes the electrochemical foundation, whereas the efflux system on the cell membrane constitutes the physiological basis of the BBB. While the BBB typically prevents neurotransmitters from the gut from directly accessing the brain, there is an exception for gamma-aminobutyric acid due to the presence of GABA transporters in the BBB. Despite this limited direct access, gut neurotransmitters can still indirectly impact the brain by affecting the Enteric Nervous System [22,23].

The median eminence and the arcuate nucleus, which are part of the hypothalamic nuclei, are uniquely situated and designed to communicate with the peripheral systems. These regions help release hormones and neurotransmitters made in the brain to the portal and cerebrospinal fluid, thereby reaching the autonomic nervous system without compromising the BBB. For example, corticotropin-releasing factor (CRF), a peptide hormone that regulates the release of glucocorticoids from the adrenal glands after the initiation of a stress response, crosses the BBB rapidly and effectively. CRF, synthesized in the hypothalamic paraventricular nuclear, is rapidly transported and stored in the median eminence from where it reaches the portal circulation after crossing the BBB to act on the anterior pituitary to release ACTH, which in turn acts on the adrenals to release glucocorticoids and epinephrine.

The intestinal barrier is another method by which the gut–brain axis is altered. The intestinal barrier comprises of two layers: a foundational single layer of epithelial cells linked by tight junctions and a mucus layer. The mucus layer undergoes changes in thickness and composition over time and includes secretory IgA and antimicrobial peptides [31,32]. When resident immune cells encounter microbial products, pattern recognition receptors distributed across the GI mucosa can trigger increased antimicrobial defenses, provoke intestinal inflammation, and potentially induce immunologic tolerance [33,34]. Changes in intestinal barrier integrity allow for the gut microbiome to access the gut milieu and alter GI function. Epithelial tight junction integrity is key to maintaining the intestinal barrier. Stress alters the expression of occludin, ZO-1, and other tight junction proteins [34,35], along with the secretion of CRF, which in turn influences gut permeability [33].

Alterations in gut microbiota can elevate the presence of detrimental compounds like p-cresol, potentially jeopardizing the integrity of the BBB [36,37]. Significantly elevated levels of p-cresol in the prefrontal cortex of mice susceptible to anxiety-like phenotypes have been reported [38]. Additionally, the gut-generated metabolite 4-ethyl phenyl sulfate influences brain activity and induces behaviors resembling anxiety [37]. Following the introduction of Bacteroides fragilis, the gut microbiota produces decreased amounts of neurotoxic metabolites, including 4-ethyl phenyl-sulfate, serum glycolate, and imidazole propionate. This leads to enhanced gut permeability and a reduction in behaviors associated with anxiety [39,40].

### 3.3. Lessons from Germ-Free Mice

Mice without natural gut microbiome are known as germ-free (GF), axenic, or gnotobiotic mice. Axenic mice are generated by hysterectomy rederivation, whereas gnotobiotic mice are axenic mice that have been inoculated with a cocktail of known one or more non-pathogenic (defined flora). Both axenic and gnobiotic mice must be maintained in isolators under very strict handling protocols to minimize exposure and keep them germ-free. In contrast, mice used routinely are specific-pathogen-free (SPF) mice. Germ-free mice have been critical to understanding many physiological processes. Lack of microbiome in utero and at birth results in compromised development of the neuro, immune, and GI systems, amongst others. GF mice have heightened hypothalamic-pituitary responses and display more anxiety- and depression-like behaviors than SPF mice. If GF mice are colonized early in life, several of the developmental and behavioral defects can be reversed, suggesting a critical window when a healthy microbiome is essential for development. GF mice and their progeny display altered BBB permeability as compared to SPF mice [41]. Decreased expression of tight junction proteins, namely occluding and claudin-5, responsible for regulating the BBB, leading to compromised BBB integrity. Transferring bacteria that produce SCFA or fecal matter transfers from SPF mice to GF mice helps improve BBB integrity [42,43]. Having fewer SCFA-producing bacteria and using sodium butyrate through oral gavage can also reduce permeability [37]. SCFAs seem to be crucial in developing and maintaining the BBB [44]. A mechanism through which SCFAs benefit the BBB includes their role as inhibitors of histone deacetylases, thereby influencing gene expression networks [44]. Another mechanism is the SCFA-GPR41/GPR43 signaling at the BBB that plays a role in regulating tight junctions, inflammation, and metabolism, all of which contribute to the maintenance of BBB integrity [39]. Dysregulation of SCFA signaling or alterations in gut microbiota composition can potentially disrupt these processes and compromise BBB function [44].

Intestinal commensal bacteria are known to stimulate IgA production in the small intestine by developing GALT [45,46]. However, their role in modulating the immune system in the large intestine, where most of these bacteria reside, has been unclear. Observations revealed that IgA production in the large intestine of germ-free mice was significantly lower compared to conventionally raised mice, mirroring previous findings in the small intestine [47,48]. A study conducted by Yanagibashi et al. reveals that *B. acidifaciens*, a predominant bacterium in the large intestine, boosts IgA production by increasing IgA+ B cells and activating B cells in cecal patches more effectively than in the small intestine [43]. New subsets of Th cells, like Th9 cells producing IL-9 and Th22 cells producing IL-22, along with follicular helper T-cells and emerging types of Treg cells, are now implicated in the development of inflammatory bowel disease as they alter the gut homeostasis through the release of those inflammatory cytokines [44].

### 3.4. The Vagus Nerve

The vagus nerve (cranial nerve X) contains both afferent/sensory and efferent/motor fibers. The afferent vagal fibers are responsive to diverse stimuli such as mechanoreceptors, gut peptides, neurotransmitters, and microbial metabolites and relay these peripheral signals to the various brain regions such as the nucleus tractus solitarius and the limbic system (Figure 1). The nodose ganglia harbor the vagal afferent neurons and sense responses mediated by a variety of chemosensitive and other receptors within the gut [49]. Indeed, following the brain, the gut possesses the highest concentration of nerve endings [49]. The efferent vagal fibers projecting from the nucleus tractus solitarius regulate intestinal mobility and secretion of pancreatic and adrenal glands’ endocrine activity. The vagus nerve thus plays an important role in facilitating the bi-directionality of the gut–brain axis.

#### 3.4.1. Role of the Vagus Nerve and Depression in Other Chronic Illnesses

In a historical study, partial removal of the stomach or gastrectomy leads to a high occurrence (43%) of psychiatric symptoms [50]. Full vagotomy has been applied in the treatment of severe obesity [51], leading to postoperative cognitive symptoms, primarily characterized by unexplained fatigue and changes in the perceptions of taste [52]. Obese individuals have lower microbial diversity compared to those of normal weight, leading to a distinct pool of microbial metabolites that differently affect energy homeostasis and GLP-1 secretion [53]. Additionally, vagus nerve responsiveness seems to be impaired in obesity. The microbiota-gut–brain axis might play a role in insulin resistance through not only the vagus nerve but also its close association with the gut microbiota. In humans, as well as in mouse models, transplanting the gut microbiome from healthy individuals into patients with metabolic syndrome has been shown to enhance the insulin sensitivity of the recipients [54]. Vagal afferents can be indirectly activated following the stimulation and release of neuroactive mediators from enteroendocrine cells or gut-associated lymphoid tissue. Beyond their potential role in regulating gut functions such as motility and secretion and in activating immune responses, the gut microbiota is also well-documented to influence “higher” CNS functions, including mood, stress-related psychiatric conditions, and memory [55,56]. A study by Kotackova et al. involved 18 participants with severe obesity who underwent bariatric surgery and completed three visits: one before the surgery and two after [57]. The study measured levels of C-reactive protein, adipose tissue, visceral fat, and depressive symptoms. The results demonstrated a reduction in depressive symptoms and improved verbal reasoning, which correspond with the post-surgical reduction in both visceral and subcutaneous fat, as well as C-reactive protein levels. These findings suggest that the improvement in mental health following bariatric surgery may be due to the decrease in systemic inflammation associated with reduced adiposity [57]. However, the consensus on the effectiveness of bariatric surgery in reducing mental health symptoms is mixed [58]. While some studies report positive outcomes over three-year follow-up [59], others indicate an increase in de novo depression and anxiety post-surgery [60], and some find no significant correlation [61].

A pilot study investigating the use of vagus nerve stimulation in 60 patients with treatment-resistant depression revealed notable clinical enhancements in 30–37% of individuals, alongside a strong tolerance to the treatment [61,62]. Soon after this study, the FDA approved the use of vagus nerve stimulation as routine treatment-resistant depression management [61,62]. In both human and rat studies, the effect of obesity and exposure to a high-fat diet on the ability of the vagus nerve to respond to gastrointestinal neuropeptides is altered [63,64]. In patients with rheumatoid arthritis, it was found that it was due to diminished vagus nerve activity, possibly from elevated serum HMGB1 protein [63,64]. Furthermore, those with cardiovascular diseases also had elevated inflammatory markers such as IL-6 and C-reactive protein. These studies indicate the crucial role of the vagus nerve in regulating immune responses and inflammation [63,64]. To date, however, supporting evidence in humans is still limited.

#### 3.4.2. Role of the Vagus Nerve in Animal Models of Anxiety- and Depression-like Behaviors

In rodents, injecting LPS into the peritoneum is a common method to mimic sickness behavior linked to both peripheral and central inflammatory responses [65]. LPS, found in the outer membrane of Gram-negative bacteria, strongly activates the human immune system. Numerous studies have observed that vagotomy diminishes the impact of LPS on various aspects of sickness behavior, such as decreased general activity two hours post-injection [65], alterations in social behavior [66], and changes in food-motivated behavior [66]. Despite this, vagotomy does not influence the levels of peripheral IL-1B, which increase after LPS treatment [67]. On the contrary, vagotomy does not prevent the LPS-induced rise in brain IL-1B in rats [68]. This suggests that while vagotomy has an impact on certain aspects of sickness behavior associated with peripheral inflammation, it does not prevent the increase in IL-1B in the brain induced by LPS in rats, which implies that the communication between the gut and the brain, often referred to as the gut–brain bidirectional pathway, is not solely dependent on the vagus nerve. Other pathways or mechanisms may be involved in transmitting signals from the gut to the brain, and the vagus nerve may not be the exclusive route for conveying gastrointestinal perturbations to the central nervous system in response to LPS-induced inflammation.

## 4. Mood Disorders, Anxiety and Depression—The Role of Microbiota on Pathophysiology and Symptoms

### 4.1. Mood Disorders Overview

Understanding the pathophysiology of mood disorders is a challenging task, as studying animal models can be quite different from how humans behave and perceive the world. Nevertheless, several studies have revealed little data on how mood disorders develop. There are several hypotheses for mood disorders depicting a complex system where multiple mechanisms are involved, including dysfunction of different signaling pathway systems within neurons, including glutaminergic [69,70], norepinephrine [71], protein kinase C signaling [72], or our innate immune system [73]. The overarching mechanism, however, lies within the deficits of the reward system in the brain. Besides bacteria, ancient human endogenous retroviral DNA present in the human genome is also linked to susceptibility to psychiatric disorders such as schizophrenia, bipolar disorder, and depression [74]. More than half the human genome consists of transposable elements such as Alu, LINE, and SVA. Transposable elements, through cis-regulation of genes, are thought to regulate gene function and contribute to neurocognitive diseases, such as schizophrenia [75]. The external environment also plays an important role in mental health; for example, the experience of childhood trauma is associated with the risk of developing mood disorders and the severity of the disorder [76].

### 4.2. Anxiety, Depression, and Post-Traumatic Stress Disorder—Pathophysiology and Microbiota

Anxiety disorders are commonly comorbid with eating disorders [77,78]. The microbiome significantly influences the early development of the hypothalamic-pituitary-adrenal axis, a key element in the stress response, as well as its responsiveness later in adulthood. Recent research indicates that prenatal stress in mice can lead to changes in the microbiome, levels of cytokines, and decreased brain-derived neurotrophic factors in offspring [77,78]. This suggests a connection between the microbiome during early stages of life and behavioral changes in adulthood [79]. Stress reduces the levels of certain crucial intestinal epithelial tight junction proteins, such as claudin 1, leading to compromised integrity of the gut epithelium [80]. Consequently, this alteration affects gut motility, secretions, and mucin production [81]. These modifications in the environment where resident bacteria reside can induce shifts in microbial composition and intestinal permeability [82,83]. Notably, this may facilitate the translocation of specific gram-negative bacteria and allergens into the bloodstream, initiating inflammasome signaling and neuroinflammatory responses.

Depression is a debilitating disease worldwide, impacting well-being, social and occupational functioning, suicide, and mortality [84]. Understanding its pathophysiology is imperative in order to develop better care. In England alone, the use of antidepressants has been steadily increasing, with prescriptions doubling approximately every decade. In 2015, over 61 million prescriptions were issued. By 2023, this number had risen to 86 million [85,86]. The increase in numbers could be due to the growing prevalence of the disease or the introduction of novel drugs. It may also reflect that the burden of the disease is growing, and the treatment regimen for depression is far from achieving its goal of curing depression.

While the microbiome has been linked to the pathophysiology of depression, current data only indicates correlation rather than causation [87]. These associations pose a significant challenge, and numerous studies have highlighted that examining the exact composition of the microbiota will not yield valuable insights into understanding the impact of our microbiome. Instead, focusing on the biochemical pathways and composition provided by the microbiome appears to be more important [88]. The gut microbiota facilitates communication between the GI tract and the central nervous system through biochemical signaling pathways. This involves regulating the levels of circulating serotonin, kynurenine, tryptophan, and SCFA.

These metabolites also play crucial roles in gut function. Specifically, butyrate, an SCFA, serves as the primary energy source for colon cells and helps defend against colorectal cancer and inflammation by inhibiting histone deacetylases [89]. While all SCFAs inhibit histone deacetylases, butyrate uniquely affects specific receptors such as GPR41/FFAR3, GPR43/FFAR2, and hydroxycarboxylic acid receptor GPR109A/HCAR2, potentially reducing the production of inflammatory cytokines [90]. SCFAs like butyrate play a crucial role in maintaining gut barrier integrity and influence the central nervous system by modulating the expression of brain-derived neurotrophic factors. SCFAs have been identified as significant factors in psychiatric disorders, with lower levels observed in depression [91]. Additionally, SCFAs are essential for preserving gut barrier function. Dietary fiber, such as butyrate, priopionate, and heptanoate, is metabolized by gut bacteria to yield SCFA. Propionate suppresses cytokines IL-1β and IL-18 in lung macrophages to change their immune tone and provide protection from injury in mice. Propionate also metabolically reprograms macrophages, switching them from glycolysis to oxidative phosphorylation after the LPS challenge. However, a high-fiber-rich diet also significantly reduces bacterial diversity [92]. Thus, metabolites produced by the gut bacteria modulate the function of cells and organs throughout the body.

Essential and non-essential amino acids, as well as biogenic amines, not only serve as neurotransmitters (e.g., glutamate, glycine, serine), they are building blocks of proteins and peptide hormones involved in a plethora of functions that include membrane stabilization, neurotransmission, and neuroimmune modulation. Tryptophan metabolite, kynurenine, is associated with stress exposure, neuronal cell death, glutamate transmission, and neuroinflammation in patients with major depression [93]. Changes in kynurenine levels, a BBB permeable amino acid metabolite, also impact neuronal function. In patients with major depression, hippocampal and amygdalar volumes correlate with the ratio of kynurenic acid/quinolinic acid derived from tryptophan [93]. In astrocytes, kynurenic acid acts as a NMDA/nicotinic receptor antagonist, whereas quinolinic acid acts as an NMDA receptor agonist predominantly in both microglia/ϕ phages. Thus, alterations in tryptophan, an essential amino acid derived from the actions of gut microbiota, not only serve as a precursor of serotonin but several other key derivates that influence neuronal function.

Interestingly, a meta-analysis of studies (*n* = 2643 patients and *n* = 2336 controls) on the microbiome in various psychiatric disorders such as MDD, psychosis, schizophrenia, bipolar disorder, anxiety, and anorexia revealed no consistent pattern in the diversity of bacteria that can be specifically linked to a particular psychiatric disorder. Additionally, individuals with these psychiatric conditions did not show a decrease in microbial diversity, often referred to as “richness”. The only notable finding was a reduction in butyrate-producing bacteria in patients with MDD, schizophrenia, and anxiety [94].

Posttraumatic stress disorder (PTSD) is a common, debilitating condition that may develop after exposure to an actual or threatened death, serious injury, trauma, or sexual violence. Symptoms of PTSD include (i) intrusive memories-recurring memories, reliving traumatic events; (ii) avoidance; (iii) negative changes in thinking and mood (hopelessness, lack of emotions), and (iv) changes in physical and emotional reactions (trouble sleeping, self-destructive behavior, aggressive behavior, overwhelming guilt or shame). Currently, the microbiome is not a contributing factor for PTSD in the Diagnostic and Statistical Manual-5. Several studies have suggested that women may be particularly vulnerable as studies have reported that women develop PTSD at twice the rate of men, despite greater trauma exposure in men [94,95,96]. It was previously found that individuals suffering from PTSD had higher levels of branched-chain and sulfur-containing amino acids, unsaturated fatty acids, and decreased levels of indoles and cyclic amino acids compared with individuals without PTSD. Among women, PTSD was associated with lower serine levels [97]. Serine serves as a neurotransmitter, as well as a precursor for the synthesis of glycine, cysteine, and 2-aminobutyric acid, butyrate, and is synthesized directly from glucose. In the glia and neurons, glycine induces the release of serine, a co-agonist for NMDA receptors. Together, these two amino acids regulate long-term potentiation [98] and are critical for fear extinction consolidation [99]. Better elucidation of the processes is the next stage of understanding our gut microbiome with depression.

### 4.3. Depression—Symptoms and Microbiota

Symptoms of depression typically include disturbed sleep, interest, feelings of guilt, low energy, decreased concentration, appetite changes, psychomotor retardation, and suicidal behavior [100]. Potential research avenues could involve investigating the mechanisms through which the pathophysiology of depression contributes to these symptoms and exploring how such insights could be leveraged to alleviate them. Specifically, this review will go over how mice studies have revealed a correlation between anxiety and depression with gut microbiota. Depression induced by stress in rodents has been linked to unusual levels of short-chain fatty acids and other metabolites associated with gut microbiota, including alanine, isoleucine, L-threonine, serine, and tyrosine [101]. These alterations may be connected to changes in serotonin levels in the brain and the manifestation of depressive-like behaviors [101,102]. As per Bravo et al., the effects of *Lactobacillus rhamnosus* on brain chemistry and depressive-like characteristics, as well as stress-induced depression-like traits in mice, were averted by vagotomy [103]. Another study showed that after lipopolysaccharide treatment, subdiaphragmatic vagotomy demonstrated a reduction in depression-like phenotypes, pro-inflammatory cytokine levels, synaptic protein expression, and abnormal gut microbiota composition in mice [67].

### 4.4. GI and Non-GI Symptoms in Anxiety and Depression and Microbiota

Individuals experiencing anxiety and depression commonly exhibit disrupted gastrointestinal symptoms, including constipation, abdominal discomfort, vomiting, nausea, and bloating [104]. Anxious patients often display symptoms resembling those of irritable bowel syndrome (IBS) or psychological distress [105]. The connection between depression and gastrointestinal symptoms is commonly linked to two primary factors: the challenges posed by chronic illness and disruptions in the brain-gut axis [106]. This link between early-life stress, mood, and gastrointestinal disorders stands out from other chronic conditions, given the substantial interplay between the central nervous system and the gastrointestinal tract [107]. Studies have indicated that corticotropin-releasing factor (CRF) in the brain and the gut [108] plays a crucial role in mediating the connection between emotional distress and alterations in both upper and lower GI motor function [109,110]. Within functional GI disorders like IBS, functional dyspepsia, and chronic constipation or diarrhea, the malfunction of the autonomic nervous system, which directly influences CRF, could contribute to changes in bowel habits and gastric emptying [111]. Transient gastric irritation and subsequent activation of the gastric mast cells in the gut also provide feedback to the hypothalamic-pituitary axis to modulate hypothalamic CRF and CRF_1_ receptor expression as well as anxiety- and depression-like behaviors in rats [112,113]. Gastric mast cells also modulate vagal activity, which in turn modulates Crh signaling in the limbic region and enhances pain perception [114]. These studies demonstrate the bi-directionality of the gut–brain axis in gut–brain disorders.

Likewise, depression is linked to heightened activity in CRF neuronal pathways [115], and CRF receptors have been proposed as potential targets for treating both depression and GI disorders [116]. CRF receptors are found in exosomes of IBS patients and correlate with IBS disease severity [117]. It is possible that consistent activation of the stress pathways mentioned above may lead to dysfunction in the brain-gut axis, making anxious and depressed patients more susceptible to symptoms such as chronic diarrhea or chronic constipation [118]. There are, of course, other reasons as well; for example, in an Iranian study, a higher prevalence of functional constipation was observed in women and individuals with lower physical activity levels [106]. In the study sample, the prevalence of depression and constipation was 28.6% and 23.9%, respectively [106]. Psychiatric conditions, including depression, might be linked to constipation either through the impact of the disease itself or other related factors [106]. Individuals experiencing depression and other psychiatric disorders commonly adopt unhealthy lifestyles, including inadequate dietary choices, insufficient fluid intake [119], and a sedentary lifestyle, as noted in the current study population.

An increasing number of research indicates the close association between the occurrence of functional constipation and an imbalance in gut microbiota [120,121]. In individuals with functional constipation, there is a decrease in the abundance of *Bacteroides*, *Rosiella*, and *Faecococcus* in the intestines, whereas pathogenic bacteria with genes linked to the methanogenic pathways, hydrogen production, and glycerol, such as *Escherichia coli* and fungi are notably increased [122]. One study proposed that depressed individuals exhibited higher levels of *Clostridium* and *Actinomycetes* in the intestinal tract compared to healthy individuals [123]. Another study discovered that depressed individuals had reduced levels of *Bacteroidetes* and *Lactobacillus* compared to their healthy counterparts [124]. A fecal transplant intervention trial revealed that increasing gut microbiota diversity led to improvements in depression and anxiety symptoms among patients with functional constipation [125]. Liang et al. conducted a cross-sectional study examining the microbiota flora in depressed patients with functional constipation. Their analysis revealed that elderly individuals with functional constipation and depressive symptoms showed decreased relative abundances of certain microbiota genera (g), including g_*Candidatus-Solibacter*, g_*Pseudoramibacter*-*Eubacterium*, g_*Peptoniphilus*, and g_*Geobacter* [126]. The rationale behind this observation may stem from g_*Candidatus’s* close association with depression, attributed to purine metabolism and fatty acid metabolism [127]. This suggests that g_*Candidatus* holds the potential to address both depression and constipation symptoms. Furthermore, the decreased abundance of g_*Peptoniphilus* and g_*Pseudoramibacter*_*Eubacterium* was negatively correlated with higher patient health questionnaires or the PHQ-9 scores, indicating a more severe depressive state in individuals with functional constipation. The major metabolite of g_*Peptoniphilus*, acetate, has the capacity to regulate mood and establish an acidic environment conducive to promoting defecation [128]. On the other hand, g_*Pseudoramibacter*_*Eubacterium*, being a primary producer of short-chain fatty acids, serves as an energy source for intestinal cells and exerts anti-inflammatory effects in the intestinal tract through the metabolism of dietary fiber [129]. It is worth noting that studies have revealed a high abundance of *Eubacter* in constipated patients, necessitating further investigation in subsequent research [130]. There is evidence that *Clostridiales*, *Lactobacillales*, and *Bacteriodales,* which together comprise about 60% of the microbiota, show diurnal fluctuations in mice [131]. This influence on circadian rhythms may offer insights into why individuals with depression often experience symptoms like insomnia. The fluctuations in these microbial species are influenced by factors such as food intake, dietary composition, biological clock, and the sex of the host [131]. Although several mechanisms have been proposed to explain the link between depression and insomnia, no consensus has been reached thus far. *Escherichia coli* and *Enterococcus*, common residents of the intestinal tract, are known to produce serotonin (5-HT), which has been implicated in the regulation of REM sleep [132]. Moreover, since 90% of the body’s serotonin is derived from chromaffin cells in the gastrointestinal tract, and certain spore-forming bacteria can modulate its synthesis and secretion, the GI tract plays a significant role in this process [132].

## 5. Sex Differences in Mental Health: An Understudied Area

Epidemiological studies indicate sex differences in the prevalence of several mental health disorders. The prevalence of mood disorders is seen to be notably higher in females compared to males, with females being more than twice as likely to be affected [133,134]. However, very few studies have addressed the role of sex as a biological variable in mental disorders, whereas the role of gender remains unstudied.

### 5.1. Role of Steroid Hormones in Mental Health

A potential involvement of gonadal hormones in the development of anxiety and depressive disorders has been suggested. Research indicates that women are particularly susceptible to mood disturbances, anxiety, and depression during periods of hormonal fluctuations, such as puberty, menopause, and the perimenstrual and post-partum phases [135,136]. There have been several studies that pointed toward a correlation between testosterone levels and the incidence of mental health disorders. However, there are mixed results that also indicate no benefit of administering testosterone to androgen-deficient patients [137,138]. Overall, the majority consensus, however, is that there is a benefit of testosterone in maintaining mental health in those with lower-than-normal levels [139]. The connection between testosterone levels and anxiety disorders, as well as MDD, is evident in males suffering from hypogonadism. This condition, characterized by reduced gonadal function and consequent low testosterone levels, is associated with a notably higher prevalence of anxiety disorders and MDD compared to individuals with normal androgen levels [140,141]. The application of a small amount of testosterone in women diagnosed with treatment-resistant MDD resulted in notable improvements in depression ratings compared to subjects who received a placebo [142]. The authors say, however, that it indicates women with lower pre-treatment androgen levels may experience more positive effects from testosterone administration on depression compared to those with higher levels. Additionally, the mean pre-treatment androgen levels in their group of women with treatment-resistant depression were at the lower end of the normal range, suggesting a potential role of relative androgen deficiency in treatment resistance.

Confirmatory randomized, placebo-controlled studies are needed to validate these results and assess the tolerability of this treatment [142]. Studies, including Schmidt et al. and Bloch et al., have attempted to explain the reason for the sex differences, saying that some subgroups of women undergo typical reproductive hormone changes but exhibit a sub-optimal central nervous system response, resulting in negative affect and maladaptive behaviors [143,144].

### 5.2. Post-Partum Depression, Pregnancy and the Role of Gonadal Steroid Hormones on Microbiota

Postpartum depression is common, with a global prevalence ranging from 4% to 25% [145]. It is one of the most frequent complications and is associated with several negative consequences for the woman, her child, and those around her. It is widely speculated that postpartum depression is partially caused by the rapid changes in reproductive hormones, including estradiol and progesterone, before and immediately after delivery [146]. While many human and animal studies indicate that shifts in these hormone levels contribute to postpartum depression, several studies have not found a significant association between hormone concentrations and postpartum depressive symptoms [146]. Regardless, the role of neuroactive steroids and GABA in the pathophysiology of postpartum depression has prompted research into synthetic neuroactive steroids and their analogs as potential treatments for the condition, such as brexanalone—a synthetic allopregnanolone [145].

Some studies attempt to explain the role of steroid hormones in normal physiology. In the forebrain and hippocampus, ovariectomy reduces brain-derived neurotrophic factor (BDNF) levels, whereas estradiol increases them [147]. BDNF levels are typically decreased by depression and stress and increased by antidepressants. Progesterone also regulates the synthesis, release, and transport of neurotransmitters [148]. In a study involving healthy women, regional cerebral blood flow was reduced in the dorsolateral prefrontal cortex, inferior parietal lobule, and posterior inferior temporal cortex during GnRH agonist-induced hypogonadism. However, the characteristic pattern of cortical activation reappeared during both estradiol and progesterone add-back [149]. In healthy humans, micromolar quantities of the glucocorticoids tetrahydrocorticosterone and tetrahydrodeoxycorticosterone are detected in bile [150,151]. The levels of these metabolites in bile are 5- to 10-fold higher in pregnant women [152]. Once these glucocorticoids are secreted into the gut, certain bacteria have the ability to chemically alter them, resulting in the production of different sets of metabolites, including progestin. There exists the possibility for neuroactive progestins to influence the activity of membrane-bound GABA and NMDA receptors present in sensory neurons accessing the gut. This could subsequently impact neurological signaling pathways within the host. The study by McCurry et al. demonstrated that gut bacteria, specifically *Gordonibacter* and *Eggerthella,* can convert tetradeoxyhydrocorticosterone into tetrahydroprogesterone [153].

## 6. Treatment of Mood Disorders

### 6.1. Current Therapy and the Need for Newer

Despite several antidepressant treatments, many patients do not experience any relief. There is no curative treatment for anxiety and depression, and nearly 30% of patients who are on antidepressants have recurrent or treatment-resistant depression [154]. Additionally, in nearly 15% of patients, psychotherapy and pharmacological therapy interventions have no effect [154]. The current approach to treating depression is contingent on its severity. In cases of moderate to severe depression, where the patient experiences significant interference with daily functioning [154], approved treatments include psychological interventions such as behavioral activation, cognitive-behavioral therapy, meditation, and interpersonal psychotherapy [154]. Pharmacotherapy options involve antidepressant drugs such as selective serotonin reuptake inhibitors, serotonin–norepinephrine reuptake inhibitors, atypical antidepressants, tricyclic antidepressants, serotonin antagonist and reuptake inhibitors, or lithium [155]. Additionally, alternative treatments include electroconvulsive therapy, deep brain stimulation, and bright light therapy. In addition to the inefficacies of antidepressant pharmacotherapy, current antidepressants have notable side effects. For instance, weight gain is a prominent adverse effect of serotonin reuptake inhibitors. A cohort study involving 294,719 individuals revealed that the use of antidepressant drugs is associated with a sustained increase in the long-term risk of experiencing a weight gain of 5% or more at the population level [156]. Consequently, an increase in weight may be linked to the exacerbation of depressive conditions [156]. More serious adverse effects include extrapyramidal symptoms, agranulocytosis, syndrome of inappropriate anti-diuretic hormone, seizures, anticholinergic symptoms, loss of libido, sedation, hypotension, galactorrhea, and QT prolongation [156]. Even in instances of mild depression, antidepressants are not the initial treatment of choice. Psychosocial interventions are favored in such cases [157]. Alongside the pharmacotherapies, natural or alternative approaches, including herbal medicines, physical activity, exercise, meditation, mindfulness, nature therapy, and music therapy, are also recommended. Additionally, maintaining a balanced diet, ensuring proper nutrition, minimizing stress, and prioritizing ample sleep are also advised [158,159,160,161].

### 6.2. Selective Serotonin Reuptake Inhibitors (SSRIs) Use and the Effect on Microbiome

There is evidence suggesting that SSRIs can induce dysbiosis in the gut microbiome, which may contribute to the side effects associated with SSRIs. It also explains the variability in adverse effect profiles among individuals and treatment resistance in some cases [162]. However, studies on this subject have yielded mixed results in both humans and mice. For example, a study by Thapa et al. involving 110 adolescents with depression (aged 15–20), 27 healthy controls, and 23 psychiatric controls on SSRIs found no significant differences in microbiome composition between the groups [163]. Interestingly, their study did show that risperidone altered the Bacteroidetes to Firmicutes ratio. In mice, the findings were contrary to these human results. SSRIs are prescribed to approximately 3% of pregnant women worldwide and are commonly used as a first-line treatment for perinatal mental illness. The impact of SSRIs on gestational stress and the microbiome is particularly intriguing. In a study with pregnant mice treated with fluoxetine and exposed to stress, the expected increase in *Bacteroidetes* species did not occur [164]. Furthermore, the treatment prevented stress-induced changes in bacterial taxa, including *Bifidobacterium*, *Bacteroides*, and *Erysipelotrichaceae*. There is a need for further research to delineate the microbiome profiles or “fingerprinting” of patients, enabling a more tailored approach to depression treatment [165]. Another intriguing aspect of gut microbiology and treatment response is the potential for these factors to cause tachyphylaxis. This suggests that slowing the treatment-associated changes in the microbiome could lead to the selection of specific resistant strains within the microbiome. Therefore, incorporating microbiome-targeted mitigation strategies should be considered [166].

### 6.3. The Placebo Effect

The placebo effect has long been known to cause amelioration of symptoms, regardless of the disease indication. In recent years, placebo has been recognized as a treatment in itself. The role of placebo in treating depression can be significant, influencing both clinical outcomes and the understanding of antidepressant therapy. In a study involving 35 individuals with pharmacologically untreated MDD, participants received either an “active” placebo pill (described as a fast-acting antidepressant) or an “inactive” placebo pill (described as having no antidepressant effects) in the first phase [167]. After one week of treatment, participants switched pills and underwent a questionnaire and a PET scan to measure µ-opioid receptor activity. Researchers found that the participants reported a decrease in depressive symptoms after taking the “active” placebo pill, which was associated with increased µ-opioid receptor activity in brain regions involved in emotion and stress regulation. Moreover, participants responded better to the “active” placebo in the first phase compared to the standard antidepressant therapy in the second phase.

This demonstrates that patients’ expectations and beliefs about treatment can significantly impact their mental health. The therapeutic relationship and the presentation of treatment can enhance the placebo effect. According to Zubieta et al., some individuals may be more responsive to the intention of treating their depression and could benefit from incorporating psychotherapies or cognitive therapies that strengthen the clinician-patient relationship alongside antidepressant medications [168]. On the contrary, a large meta-analysis (*n* = 3228) of over 50 randomized controlled trials found that while the placebo effect on treatment-resistant depression was statistically insignificant, it was still widely observed [169]. The findings of this study suggest that establishing a benchmark for the placebo effect size can aid in interpreting the results of both past and future clinical trials.

### 6.4. The World of Prebiotics, Probiotics, Postbiotics and Synbiotics

Probiotics literally means “for life” and are used to define living non-pathogenic organisms or microorganisms that do not cause disease, which have positive effects on their hosts, contributing to various beneficial outcomes [170]. Some of the popularly used probiotics are *Lactobacillus*, *Bacillus*, *Enterococcus* species [171], and *Bifidobacterium* [172]. Prebiotics primarily consist of non-digestible food ingredients, positively influencing the health of the host by selectively promoting the growth and/or activity of specific groups of microorganisms in the colon. Typically, these include short-chain carbohydrates or cocoa-derived flavanols [173]. Natural sources of prebiotics include tomatoes, artichokes, bananas, asparagus, berries, garlic, onions, chicory, green vegetables, legumes, as well as oats, linseed, barley, and wheat [174]. These, however, do not fit the Wang [175] selection criteria for prebiotics (Table 1). Thus, artificially synthesized prebiotics, including fructooligosaccharides, maltooligosaccharides [174], lactosaccharose and cyclodextrins [174], galactooligosaccharides [170], and lactulose [176], are used in studies. Synbiotics are a combination of prebiotics and probiotics which together produce a synergistic effect [171]. The most popular synbiotic is a combination of *Lactobacillus* or *Bifidobacterium* with fructooligosaccharides. Lastly, postbiotics are a preparation consisting of non-living microorganisms and/or their components that provide a health benefit to the host [177]. Acquisition of these can be obtained from the lysis of bacteria, enzyme extraction, solvent extraction, sonication, and heat application of various forms of postbiotics [178]. These include cellular wall fragments, enzymes, SCFA, vitamins, phenols, or exopolysaccharides [178]. There is still insufficient data regarding the benefits of postbiotics due to the heterogeneity of the classification; however, some effects that are being considered are immunomodulation [179], infection prevention [180], lipid/cholesterol metabolism [181], and antitumor/antioxidant activity [182]. The idea of using non-living microorganisms to support or maintain health is not novel. Other terms used to describe such substances include “paraprobiotics” [183,184], “heat-killed probiotics” [185,186], “metabiotics” [187], and “bacterial lysates” [188].

Potential mechanisms for mediating health effects through postbiotics are akin to those of probiotics [189] and encompass the improvement of epithelial barrier function, regulation of host-microbiota interactions, modulation of immune responses, adjustment of systemic metabolism, and signaling through the nervous system [190]. In a study addressing use in patients with irritable bowel syndrome, the usage of synbiotics proved beneficial [171]. However, due to heterogeneous results, more studies are required to address this. Probiotics have demonstrated efficacy in addressing various mental conditions, including anxiety [159,160], post-partum depression [191], stress and memory issues [192], mood and sleep disturbances [193], anger [160,161], as well as anxiety and depressive symptoms associated with schizophrenia [194,195]. A study looking at the effects of probiotics on the gut microbiome showed mixed but promising results. “OMNi-BiOTiC Stress Repair” is a commercial product that contains various strains of *Lactobacillus* and *Bifidobacterium* [196]. After administration of “OMNi-BiOTiC Stress Repair” and following up after 4 weeks, the researchers noticed a significant difference in beta diversity but no change in alpha diversity along with an increase in Coprococcus species [196]. Additionally, within the Flemish gut flora project, an extensive study on microbiome research encompassing over a thousand individuals with depression compared to healthy controls revealed a depletion of *Coprococcus* species in individuals with depression [80]. This observation is of particular significance as *Coprococcus* is recognized for its production in butyrate, a compound that increased during probiotic supplementation [197].

### 6.5. Human Studies—Pro, Pre, and Postbiotics

Probiotics—In a study by Wallace et al., participants received probiotic supplementation containing *Lactobacillus helveticus* and *Bifidobacterium longum* once daily for 8 weeks [198]. The findings revealed that daily intake of probiotics notably decreased anxiety and enhanced overall mood and anhedonia by week 4, along with improvements in sleep quality by week 8 [198]. No significant changes were observed in the levels of IL-6, IL-1β, TNF-α, and cortisol in either the probiotic or placebo groups. However, a significant reduction in kynurenine concentration and enhancement of cognitive functions were noted in the *Lactobacillus helveticus* group compared to the placebo group, subsequently leading to improved cognitive functions [199]. Although the supplementation did not alleviate symptoms of depression, it did reduce the severity from moderate to mild [198]. This study thus suggests the potential benefits of augmenting standard depression therapy with specific strains of probiotic bacteria [199]. In a study by Majeed et al., patients experiencing both irritable bowel syndrome and MDD were given a once-daily dose of probiotic *Bacillus coagulans* for 90 days. Their findings showed that supplementation with *Bacillus coagulans* significantly relieved symptoms of clinical depression [200]. Furthermore, the probiotic strain exhibited a favorable impact on sleep disturbances and reduced levels of myeloperoxidases, which are implicated in regulating the production of free radicals associated with cellular oxidative stress and thus linked to depression and certain neurodegenerative diseases [200].

Prebiotics—In a clinical trial conducted by Ghorbani et al., the effects of particular probiotics and a fructo-oligosaccharide prebiotic as an adjunct therapy to fluoxetine were investigated [201]. The study showed that participants in the probiotic and prebiotic group had notably lower Hamilton Depression Rating Scores compared to those in the placebo group, indicating the beneficial role of this symbiotic combination as a complementary treatment [201]. In another study examining the effects of probiotic, prebiotic, and postbiotic supplements as adjuncts to antidepressant therapy, it was observed that probiotic supplementation improved the Beck Depression Inventory score compared to the placebo group. However, prebiotic supplementation did not affect the Beck Depression Inventory scale results in severe depressive cases. The variation in antidepressant drugs taken by participants may have influenced this outcome, and further research should be conducted to include this limitation [202]. On the contrary, Vaghef-Mehrabany et al. found no statistically significant differences in depression symptoms between the prebiotic and placebo groups following supplementation [203]. The study indicates that although the administration of the prebiotic had minimal impact on metabolic changes, the reduction in calorie intake and subsequent weight loss had a more pronounced effect on improving the well-being of obese patients with MDD [203]. This also solidifies our understanding of the importance of lifestyle-based changes in anti-depressive therapy. Table 2 summarizes some findings and the relationship between various microbiota treatments and mental health.

Postbiotics—A study looking at the salivary cortisol levels in medical students before an examination compared to the placebo after the consumption of washed and heat-inactivated cells of *Lactobacillus gasseri* showed reduced cortisol levels [204]. Questionnaires assessing mental and physical states revealed that significantly reduced anxiety and sleep disturbance compared to placebo [189]. Additionally, fecal microbiota analyses demonstrated that diet supplementation with *L. gasseri* mitigated the stress-induced decline of *Bifidobacterium* spp. and the stress-induced elevation of *Streptococcus* spp. The precise mechanism underlying these stress-relieving effects remains unclear, warranting further investigation [205]. In another study involving 241 participants, individuals were randomized to receive either heat-killed *L. paracasei* cell powder or a placebo once daily for 12 weeks [206]. Furthermore, no adverse effects associated with postbiotic supplementation were observed during the study period [206].

**Table 2 cells-13-01436-t002:** Summary of possible adjunctive therapies to mental health disorders. * HDRS—Hamilton Depression Rating Scale.

Treatment Type	Example	Effects
Probiotics	*Lactobacillus*, *Bacillus*, *Enterococcus*, *Bifidobacterium*	Decreased anxiety and enhanced overall mood [198], improved cognitive functions [199], increased sleep [200]
Prebiotics	*Fructooligosaccharides*, *Maltooligosaccharides*, *Lactosaccharose*, *Cyclodextrins*, *Lactulose*, *Galactooligosaccharide*	Mixed results; Ghorbani et al. demonstrated low HDRS * scores [201], Vaghef et al. study showed no benefit [203],
Postbiotics	Cell free supernatant, enzymes, SCFA, vitamins, phenols, exopolysaccharides, bacterial lysates, cellular wall fragments	Decreased cortisol levels [204], mitigated stress induced decline of *Bifidobacterium* [205] immunomodulator [179], antitumor/antioxidative effects [182]
Synbiotics	*Lactobacillus*/*Bifidobacterium* plus fructooligosaccharide or psyllium or inulin	Immunomodulation, maintenance of gut microbiota [201]
Microbiota Therapeutics	Fecal matter transplants, synbiotic microbial consortia, engineered symbiotic microbes	Ameliorates gastrointestinal symptoms, better sleep [207]

### 6.6. Fecal Matter Transplants (FMT)

Microbiota therapeutics encompass various interventions, such as whole fecal microbiota transplants (FMT), symbiotic microbial consortia, or engineered symbiotic microbes [207]. The primary objective of microbiota therapeutics is to restore dysbiotic microbiota to a state of health by introducing a beneficial and balanced microbial community. A randomized controlled trial evaluated the effectiveness of oral frozen FMT capsules as an adjunct therapy in patients with MDD and discovered that depressive symptoms significantly improved four weeks after transplantation [208]. Likewise, FMT therapy for depression also ameliorates gastrointestinal symptoms and restores the balance of the gut ecosystem [209]. This suggests that there is potential for future research exploring alternative pills derived from human feces, aiming to achieve similar effects with less invasiveness and greater standardization [210]. A pilot RCT study by Green et al. explored whether FMT is feasible, acceptable, and safe, and their results were promising. In this study, 15 patients experiencing MDD were administered FMT through enemas, and results after 8 weeks were gathered to determine the outcomes, including mental health symptoms based on MADRS and Depression- Anxiety Stress Scale, quality of life measured with Assessment of Quality of Life-8 Dimensions scale, sleep measured by Pittsburgh Sleep Quality Index, and gut symptomatology measured by Gastrointestinal Symptom Rating Scale. The authors noted improvement in quality of life along with significant improvement in gastrointestinal symptoms. Additionally, no serious or severe adverse events were reported by the participants [207]. The study is not without limitations; however, as the number of participants was small, the outcomes were underpowered. Additionally, the observed mental health outcomes might have been affected by factors unrelated to the intervention, such as participants altering their psychotropic treatments during the study or contracting COVID-19 [207]. Patients with IBS-related anxiety and depression were given FMT and observed over a period of three months. The study showed that the treatment was, in fact, effective as it was able to reduce anxiety for a long period of time, thus improving the quality of life [105].

While FMT shows promise, it carries complications and risks. Colonoscopy is the preferred route for administration, as it allows delivery throughout the colon, but should be used cautiously in patients with severe colitis or ileus due to the risk of perforation. Administration via gastroduodenoscopy is also possible. However, it poses risks of aspiration small bowel bacterial overgrowth and may not reach the colon [211]. FMT has been most successful in treating recurrent *Clostridium difficile* infections, but evidence for its efficacy in IBD is still insufficient, demonstrating that its use in the clinical setting is limited and that further research is required to address this [212]. Additionally, FMT can cause mild side effects like diarrhea and vomiting, as well as severe events like infections and autoimmune diseases. Despite donor stool screening, FMT can transmit infections and affect conditions like metabolic and neuropsychiatric disorders [212].

## 7. Discussion

In this review, we explored the current comprehension of our microbiome and its bidirectional interaction with the central nervous system (Figure 1). The role of the microbiome in neurological diseases such as Autism Spectrum Disorders, Alzheimer’s, or Parkinson’s disease is beyond the scope of this review. As our health status changes, so does our microbiome, adapting to various conditions. The referenced studies illustrate the close association between different microbial species and inflammatory conditions in the body, shedding light on how mental health disorders may be elucidated within the framework of dysbiosis. It is crucial to consider various factors when interpreting the studies discussed thus far, particularly to grasp the mutual relationship between the microbiome and mental health disorders, specifically its involvement in MDD.

In 2023, the global population of individuals experiencing mental health disorders surpassed 1 billion, marking an increase from 970 million in 2019 [213,214,215]. Unfortunately, these numbers are rather an underestimation as the burden of mental disorders is likely due to insufficient recognition of complex interactions between mental illness and other health conditions [216]. The incidence of newly diagnosed mental disorders increased at an alarming rate during the COVID-19 pandemic [217,218], but the effect of the pandemic on mental health is a topic in itself and beyond the scope of this review. While social media platforms have significantly contributed to an increase in mental health disorders amongst teens and young adults, especially girls, social media and public campaigns have also facilitated more open conversations, thereby allowing increased awareness and helping reduce the stigma associated with mental health disorders.

When observing the impact of probiotics in adjunctive treatment, several of these studies lacked gut microbiota profiling of patients before and after probiotic use [91]. Furthermore, there are discrepancies among these studies regarding the strains used and the duration of treatment, which ranged from 6 to 13 weeks [219]. Three out of the five studies utilized combinations of *Lactobacillus* and *Bifidobacterium* species [142,162,163], whereas the remaining two studies focused on single strains such as *Bifidobacterium lactis* [220] and *Bacillus coagulans* [200]. Due to the limited number of studies, definitive conclusions regarding the optimal strain combinations and treatment duration cannot be made. However, long-term probiotic supplementation may hold some promise, as probiotics are not detectable in stool 1–4 weeks after consumption is ceased [219,220]. For instance, in Pinto-Sanchez’s study, depression scores remained significantly improved compared to baseline during the follow-up period (4 weeks after the end of the probiotic intervention) but began to rise again thereafter [221].

Further research is necessary to explore the interplay between dysbiosis and depression development, as well as the interactions between drugs and the microbiota, in order to understand the causal relationships among dysbiosis, depression, and treatment. Although preclinical and clinical evidence suggests that microbiota-based therapies can lead to depression remission, their mechanisms of action, the key microbiota involved, and their interactions remain poorly understood [222]. According to Liu et al., establishing a disease-based gut microbial biobank through bacterial isolation is essential, along with utilizing high-throughput sequencing, multi-omics approaches, and microbial culture techniques to isolate pathogenic and beneficial strains associated with depression. Identifying the exact functions of these strains and uncovering the underlying [223] mechanisms of the gut–brain axis are crucial steps toward building a comprehensive disease-based strain resource database [107]. Examining potential confounding effects is crucial for advancing microbiota-based diagnostics and therapeutics [223].

Majeed et al. recommend Additional prospective, larger-scale trials with extended follow-up periods to establish the underlying mechanisms and conduct a detailed assessment of the therapeutic effects of *B. coagulans* supplementation in managing MDD in patients with IBS [200]. Barandouzi et al. suggest that the conflicting findings regarding microbial diversity and abundance at different taxonomic levels among individuals with depression suggest the presence of confounding factors [124]. These are variations in study populations, including factors such as age, diet, weight, geographic location, host genetics, and behavioral aspects that can impact the composition of the gut microbiome. Another important factor proposed by Barandouzi et al. when comparing studies is the fact that while the majority of studies employed 16S rRNA sequencing for bacterial analysis, one study opted for phylogenetic analysis of bacterial peptides [124]. This divergence in methodology may introduce challenges in comparing findings across studies. Furthermore, there were variations in analytical approaches among studies, including differences in targeted regions of the bacterial DNA, thresholds for clustering operational taxonomic units, and criteria for diagnosing depression. Such discrepancies have the potential to impact the outcomes of the studies, as direct comparability may be compromised [124]. Differences observed at taxonomic levels imply a potential role of increased bacterial translocation and intestinal permeability in depression pathophysiology.

One drawback of fecal matter transplants is that infections can be transferred from one patient to another if the stool samples are not carefully tested for diseases [224]. Additionally, risk factors such as obesity, drug abuse, tattooing, and antibiotic use have to be considered when choosing donors for the stool to reduce complications in recipients [224]. Further research is required to allow tailoring of fecal matter transplants according to patients such that it can be used as an adjunct or even as an alternative to our current anti-depressive therapy standards [224].

Glucagon-like peptide-1 (GLP-1) receptor agonists are medications for diabetes mellitus that have a beneficial side effect of significant weight loss, but their long-term impact on weight gain and mental health remains unclear [225]. Interestingly, although it is uncommon for an antidiabetic drug to alter the microbiome, a study by Wang et al. demonstrated that treatment with liraglutide in mice led to significant changes in bacterial composition [226]. Specifically, obesity-related phylotypes such as *Erysipelotrichaceae*, *Marvinbryantia*, *Roseburia*, *Candidatus*, and *Parabacteroides* were reduced, which may contribute to the weight reduction observed with GLP-1 receptor agonists [215,216]. A more recent study in 2024 by Duan et al. investigated the effects of semaglutide on the gut microbiota of obese mice induced by a high-fat diet [227]. Certain strains like *Akkermansia*, *Faecalibaculum,* and *Allobaculum* were significantly decreased, whereas *Lachnospiraceae* and *Bacteroides* were increased after the high-fat diet. When these mice were given semaglutide, the previously lost gut flora was restored, and there was an increase in tight junction proteins in the intestinal barrier. This study demonstrated that the anti-obesity effect of semaglutide could be associated with dysbiosis, which can be alleviated by semaglutide.

## 8. Conclusions

Studies suggest that microbiota-based therapies can lead to depression remission; however, due to the lack of integrated and systems analysis, cross-comparison is often not possible. Further investigation is needed to fully understand the contribution of gut flora in mediating gut–brain crosstalk. Advancing this understanding will pave the way for innovative treatments, improving mental health care outcomes. Future research should examine the relationship between dysbiosis and depression, drug-microbiota interactions, and identify specific microbial strains involved in the gut–brain axis. Establishing a disease-based gut microbial biobank and employing high-throughput sequencing and multi-omics approaches are essential for isolating and characterizing strains. Access to raw data is essential for reproducibility and rigor. These efforts will uncover underlying mechanisms and develop microbiota-based diagnostics and therapeutics.

## Figures and Tables

**Figure 1 cells-13-01436-f001:**
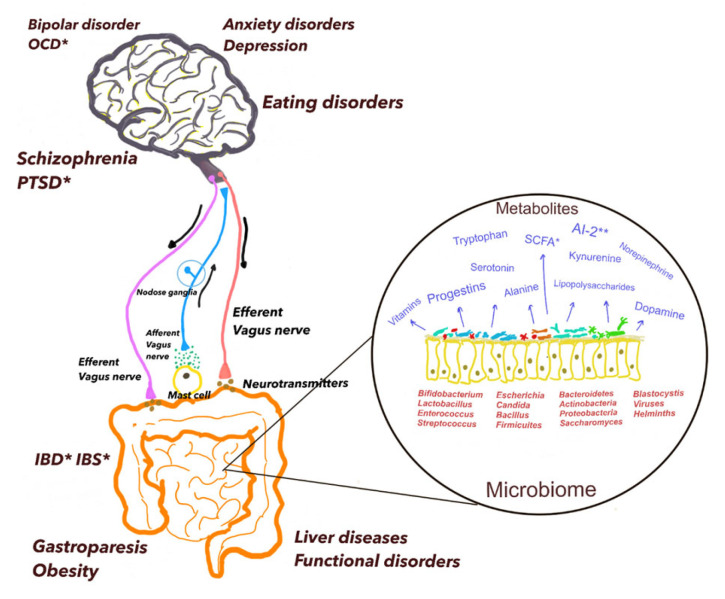
Schema illustrates the interaction of the gut and the brain and a few of its associated illnesses. OCD—Obsessive Compulsive Disorder, PTSD—Post Traumatic Stress Disorder, IBS—Irritable Bowel Syndrome, IBD—Inflammatory Bowel Disease, * SCFA—Short Chain Fatty Acids, ** Autoinducer-2. Neurotransmitters released by the vagus nerve include serotonin, acetylcholine, dopamine, and norepinephrine. The afferent fibers that pass through the nodose ganglia respond to mast cell activation as well as mechanoreceptors, gut peptides, neurotransmitters, and microbial metabolites. The afferent fibers activated by mast cells lead to the upregulation of CRF and CRF_1_ receptors, as well as the activation of the *Crh* signaling pathway in the limbic system.

**Table 1 cells-13-01436-t001:** Wang Criteria for prebiotics.

Wang Criteria
Resistance to digestion in the upper sections of the alimentary tract
Fermentation by intestinal microbiota
Beneficial effect on host’s health
Selective stimulation of growth of probiotics
Stability in various food/feed processing conditions

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
