# Peer review of "Gut-Brain Axis: Role of Microbiome, Metabolomics, Hormones, and Stress in Mental Health Disorders"

_cells, 2024, doi:10.3390/cells13171436_

Round 1

Reviewer 1 Report

Comments and Suggestions for Authors

The manuscript "Gut-Brain Axis: Role of Omics, Hormones, and Stress in Mental Health Disorders" provides a comprehensive review of the interactions between the gut microbiome and mental health, focusing on depression and anxiety disorders. It emphasizes the role of omics technologies, hormones, and stress in these processes, highlighting recent advancements and historical perspectives.

The manuscript covers a wide range of topics related to the gut-brain axis, including historical context, modern research findings, and the impact of various factors on mental health.

With that, there are several points to be improved.

Dear authors, please consider the foregoing points:

1.       The abstract could be more concise, clearly stating the main findings and significance. The introduction section is somewhat repetitive and could benefit from a more focused narrative (as lines 63-82 are not useful, as there will be no discussion of COVID-19 and PTSD further). I can suggest to streamline the narrative to maintain reader interest.

2.       The methodology section describing the search strategy for literature is brief and lacks detail. Please expand the methodology section to include specific search terms, inclusion and exclusion criteria, and the rationale behind the selection of studies.

3.       The tables summarizing major studies and the found interrelations will be of great interest.

4.       In discussion, please expand on the potential clinical applications of microbiome research in mental health and outline specific future research questions and methodologies.

Author Response

The manuscript "Gut-Brain Axis: Role of Omics, Hormones, and Stress in Mental Health Disorders" provides a comprehensive review of the interactions between the gut microbiome and mental health, focusing on depression and anxiety disorders. It emphasizes the role of omics technologies, hormones, and stress in these processes, highlighting recent advancements and historical perspectives.

The manuscript covers a wide range of topics related to the gut-brain axis, including historical context, modern research findings, and the impact of various factors on mental health.

With that, there are several points to be improved. 

Response. We thank the reviewer for comments.

Dear authors, please consider the foregoing points:

  1. The abstract could be more concise, clearly stating the main findings and significance. The introduction section is somewhat repetitive and could benefit from a more focused narrative (as lines 63-82 are not useful, as there will be no discussion of COVID-19 and PTSD further). I can suggest to streamline the narrative to maintain reader interest.

Response. Thank you for the suggestion. We have added text to the abstract to include main findings and significance. We have done our best to eliminate repetition in the introduction. We realize that there is no further discussion of COVID-19 and PTSD, but this is a topic in itself and deserves a dedicated review. Given that COVID-19 is recognized as one of the biggest risk factors for mental health, not mentioning it would be remiss. We have moved this aspect to discussion section, where it fits better with the flow and future topics. We do discuss changed omics in PTSD patients. We have revised our discussion and moved some text around and hope it is now more streamlined.

  1. The methodology section describing the search strategy for literature is brief and lacks detail. Please expand the methodology section to include specific search terms, inclusion and exclusion criteria, and the rationale behind the selection of studies.

Response. Review articles usually do not include methodology. However, we have expanded this section as requested.

  1. The tables summarizing major studies and the found interrelations will be of great interest.

Response. Table 2 summarizes major studies.

  1. In discussion, please expand on the potential clinical applications of microbiome research in mental health and outline specific future research questions and methodologies.

Response. As requested by the editor, a conclusion section has been added to discuss these aspects.

Reviewer 2 Report

Comments and Suggestions for Authors

From the numerous studies examined by the authors, it emerged that there are discrepancies among these studies regarding the strains used and the duration of treatment, perhaps due to the number of combinations examined. In Some studies they utilized combinations of Lactobacillus and Bifidobacterium species, while the remaining studies focused on single strains such as Bifidobacterium longum and Bacillus coagulans. Therefore due to the limited number of studies, definitive conclusions regarding the optimal strain combinations and treatment duration cannot be made. Which suggests in other studies the construction of a gut microbial biobank through bacterial isolation is essential along with utilizing high-throughput sequencing, multi-omics approaches, and microbial culture techniques to isolate pathogenic and beneficial strains associated with depression. Identifying the exact functions of these strains and uncovering the underlying mechanisms of the gut-brain axis are crucial steps toward building a comprehensive disease-based strain resource database. Another important factor analyzed is that when comparing studies, it is that fact that while the majority of studies employed 16S rRNA sequencing for bacterial analysis, one study opted for phylogenetic analysis of bacterial peptides. And even the less experienced will realize that this divergence in methodology may introduces challenges in comparing findings across studies. Ultimately, the novel GLP-1 agonist molecule was evaluated: When these mice were given GLP-1 in particular semaglutide, the previously lost gut flora was restored, and there was an increase in tight junction proteins in the intestinal barrier. It is good to remind the authors to never transcribe the commercial names of drugs, as in this case Ozempic and Wegovy, unless expressly requested by the pharmaceutical company that offers them.

Author Response

From the numerous studies examined by the authors, it emerged that there are discrepancies among these studies regarding the strains used and the duration of treatment, perhaps due to the number of combinations examined. In Some studies they utilized combinations of Lactobacillus and Bifidobacterium species, while the remaining studies focused on single strains such as Bifidobacterium longum and Bacillus coagulans. Therefore due to the limited number of studies, definitive conclusions regarding the optimal strain combinations and treatment duration cannot be made. Which suggests in other studies the construction of a gut microbial biobank through bacterial isolation is essential along with utilizing high-throughput sequencing, multi-omics approaches, and microbial culture techniques to isolate pathogenic and beneficial strains associated with depression. Identifying the exact functions of these strains and uncovering the underlying mechanisms of the gut-brain axis are crucial steps toward building a comprehensive disease-based strain resource database. Another important factor analyzed is that when comparing studies, it is that fact that while the majority of studies employed 16S rRNA sequencing for bacterial analysis, one study opted for phylogenetic analysis of bacterial peptides. And even the less experienced will realize that this divergence in methodology may introduces challenges in comparing findings across studies. Ultimately, the novel GLP-1 agonist molecule was evaluated: When these mice were given GLP-1 in particular semaglutide, the previously lost gut flora was restored, and there was an increase in tight junction proteins in the intestinal barrier. It is good to remind the authors to never transcribe the commercial names of drugs, as in this case Ozempic and Wegovy, unless expressly requested by the pharmaceutical company that offers them.

Response. We thank the reviewer for this excellent summary and comments. We have included some of these suggestions/comments in our conclusion. We apologize for the oversight in using commercial name and have replaced those names with GLP-1 receptor agonist, as suggested.